# Dietary Polyacetylenic Oxylipins Falcarinol and Falcarindiol Prevent Inflammation and Colorectal Neoplastic Transformation: A Mechanistic and Dose-Response Study in A Rat Model

**DOI:** 10.3390/nu11092223

**Published:** 2019-09-14

**Authors:** Morten Kobaek-Larsen, Gunnar Baatrup, Martine K. Notabi, Rime Bahij El-Houri, Emma Pipó-Ollé, Eva Christensen Arnspang, Lars Porskjær Christensen

**Affiliations:** 1Department of Clinical Research, University of Southern Denmark, 5000 Odense, Denmark; gunnar.baatrup@rsyd.dk; 2Department of Surgery, Odense University Hospital, 5000 Odense, Denmark; 3Department of Chemical Engineering, Biotechnology and Environmental Technology, University of Southern Denmark, 5230 Odense M, Denmark; mkno@kbm.sdu.dk (M.K.N.); rbeh@kbm.sdu.dk (R.B.E.-H.); emmapipoolle@hotmail.com (E.P.-O.); arnspang@kbm.sdu.dk (E.C.A.); 4Department of Chemistry and Bioscience, Faculty of Engineering and Science, Aalborg University, 6700 Esbjerg, Denmark; lpc@adm.aau.dk

**Keywords:** falcarinol, falcarindiol, COX-2 inhibition, aberrant crypt foci, anti-neoplastic effect, anti-inflammatory activity, colorectal cancer

## Abstract

Falcarinol (FaOH) and falcarindiol (FaDOH) are cytotoxic and anti-inflammatory polyacetylenic oxylipins, which are commonly found in the carrot family (Apiaceae). FaOH and FaDOH have previously demonstrated a chemopreventive effect on precursor lesions of colorectal cancer (CRC) in azoxymethane (AOM)-induced rats. The purpose of the present study was to elucidate possible mechanisms of action for the preventive effect of FaOH and FaDOH on colorectal precancerous lesions and to determine how this effect was dependent on dose. Gene expression studies performed by RT-qPCR of selected cancer biomarkers in tissue from biopsies of neoplastic tissue revealed that FaOH and FaDOH downregulated NF-κβ and its downstream inflammatory markers TNFα, IL-6, and COX-2. The dose-dependent anti-neoplastic effect of FaOH and FaDOH in AOM-induced rats was investigated in groups of 20 rats receiving a standard rat diet (SRD) supplemented with 0.16, 0.48, 1.4, 7 or 35 µg FaOH and FaDOH g^−1^ feed in the ratio 1:1 and 20 rats were controls receiving only SRD. Analysis of aberrant crypt foci (ACF) showed that the average number of small ACF (<7 crypts) and large ACF (>7 crypts) decreased with increasing dose of FaOH and FaDOH and that this inhibitory effect on early neoplastic formation of ACF was dose-dependent, which was also the case for the total number of macroscopic neoplasms. The CRC protective effects of apiaceous vegetables are mainly due to the inhibitory effect of FaOH and FaDOH on NF-κB and its downstream inflammatory markers, especially COX-2.

## 1. Introduction

Colorectal cancer (CRC) is a major worldwide health problem and is the third most frequent cause of cancer-related death in developed countries [1]. The growing incidence of CRC is probably associated with a modern lifestyle typified by limited physical activity, alcohol consumption and dietary changes [2]. CRC development is a multistep process, from normal epithelial cells via inflammation to aberrant crypt foci (ACF) and progressive adenoma stages, to carcinomas and then metastatic disease [3,4]. In order to reduce the incidence of CRC, effective prevention and treatment strategies need to be identified. Due to the long precancerous stage of this disease, dietary intervention may exert a favorable effect on polyp formation and/or inhibition of the transformation of adenomas to CRC. Recent findings indicate that long-term consumption of a diet rich in vegetables may prevent the development of CRC [5]. Vegetables contain a wide variety of bioactive secondary metabolites such as glucosinolates, polyphenols and/or polyacetylenic oxylipins, some of which have shown bioactivities that may contribute to their CRC protective effects [6,7,8].

Strong evidence suggests the existence of an association between the development of CRC and the expression of cyclooxygenase (COX) enzyme complex that catalyzes the conversion of arachidonic acid released from membrane phospholipids to prostaglandins. Prostaglandins sustain homeostatic functions and mediate pathogenic mechanisms, including the inflammatory response [9]. COX-1 is a constitutively expressed enzyme in most mammalian tissues and it is involved in normal physiological functions [9,10]. In contrast, COX-2 levels are normally low but are rapidly induced as an early response to growth factors, cytokines and tumor promoters associated with inflammation, cell survival, abnormal proliferation, angiogenesis, invasion, and metastasis [9]. In particular, the existence of an association between CRC and COX-2 overexpression has been established [9,10]. Chronic inflammation is associated with a greater likelihood of carcinogenesis, and in the case of CRC, a microenvironment conducive for the development of early neoplastic lesions is created with relative high concentrations of prostaglandins [11]. COX-2 expression but not COX-1 has been detected in 50% of colorectal adenomas and in up to 85% of colorectal carcinomas, and it is correlated with poor prognosis [12]. Therefore, inhibition of the expression and/or activity of COX-2 by dietary bioactive secondary metabolites may be an important target for CRC chemoprevention. This information might have prognostic importance and may help in identifying new treatment strategies. Nonsteroidal anti-inflammatory drugs (NSAIDs) are known COX inhibitors, and they have been widely used for cancer prophylaxis; also, epidemiological studies have found them to have chemopreventive potential [13]. However, several adverse effects, including gastrointestinal bleeding, asthma, hepatic, renal and cardiovascular toxicity have been associated with their long-term usage, and hence their clinical application has been limited [14]. In this context, dietary bioactive natural products have attracted a great deal of research interest because they inherently show low toxicity and a promising safety profile with no known severe side effects [15]. 

For the prevention of CRC by dietary measures, apiaceous vegetables such as carrots, celery, celeriac, fennel, parsley, and parsnip are highly interesting due to their content of the bioactive polyacetylenic oxylipins falcarinol (FaOH) and falcarindiol (FaDOH) (Figure 1) [16,17,18]. FaOH and FaDOH have shown many interesting bioactivities, including anti-inflammatory, anti-platelet-aggregatory, anti-diabetic, and cytotoxic activity as well as an anti-neoplastic effect [17,18,19,20,21,22,23,24,25,26,27,28,29,30,31,32]. From in vitro studies, it is clear that FaOH is more cytotoxic than FaDOH and that their cytotoxicity depends on the cell lines [17,24,27,28,29,30]. In addition, it has been demonstrated that FaOH inhibits the growth of the human epithelial colorectal adenocarcinoma cell line Caco-2 in vitro, and that this effect is enhanced synergistically when combined with FaDOH in ratios of 1:1, 1:5 or 1:10 [27]. Furthermore, it has been shown that FaOH and FaDOH can lead to cell cycle arrest and apoptosis of cancer cells [25,33]. The exact mechanisms of action for the anti-proliferative effect of FaOH and FaDOH in adenoma/cancer cells are unknown, but may be due to their alkylating properties, which can lead to the inhibition of pro-inflammatory markers, enzymes, and inflammatory transcription factors via covalent alkylation [30,34]. The synergistic anti-proliferative effect of FaOH and FaDOH also seem to exist in vivo as recently demonstrated in a cancer primed rat model for CRC where a diet containing 7 μg FaOH and 7 μg FaDOH g^−1^ feed was shown to significantly inhibit neoplastic transformations in the colon epithelium [32]. The amounts of FaOH and FaDOH used in this rat study correspond to a daily human intake of 250–300 g carrots and indicates that apiaceous vegetables could have a preventive effect against CRC.

FaDOH is an effective inhibitor of COX-1 and COX-2 in vitro, whereas the COX inhibitory activity of FaOH is less pronounced [21,29,35]. In addition, FaOH and FaDOH have been shown to inhibit the formation of the pro-inflammatory cytokines interleukin (IL)-6, IL-1β and tumor necrosis factor-α (TNFα), and hence, the activation of their upstream nuclear factor kappa-light-chain-enhancer of activated B cells (NF-κB) signaling pathway [19]. This signaling pathway is crucial for neoplastic transformation and promotion [36]. FaOH and FaDOH are also strong inhibitors of lipoxygenases that are involved in tumor-progression and activation of NF-κB [20,21]. Thus, we hypothesized that a likely mechanism of action for the preventive effect of FaOH and FaDOH on colorectal precancerous lesions is mainly due to their anti-inflammatory activity and that this effect is dose-dependent. 

In this study, we show that FaOH and FaDOH have an inhibitory effect on certain inflammatory markers in neoplastic lesions and that a possible mechanism of action in relation to CRC prophylactics could be as selective COX-2 inhibitors. Furthermore, we demonstrate that FaOH and FaDOH prevent the development of early neoplastic ACFs and neoplastic polyp lesions in the colorectal intestine in a dose-response relationship. 

## 2. Materials and Methods 

### 2.1. Animals

All animal experiments were approved by the central Animal Experimentation Inspectorate in Denmark (License no. 2015-15-0201-00708). Male rats from the F344 strain with a certified health report were purchased from Charles River. The animals were 5 weeks old at the time of arrival. After 1 week of acclimatization, rats were separated into two groups and fed a standard rat diet (SRD) (Altromin 1321, Brogaarden, Lynge, Denmark) or SRD supplemented with FaOH and FaDOH as described in Section 2.2. The rats were fed on the different diets for 2 weeks before the first injection of azoxymethane (AOM) at the age of 8 weeks. All animals were housed as described in earlier studies [31,32].

### 2.2. Rat Diets and Design of Rat Feeding Experiments

The purified FaOH and FaDOH was added in a 1:1 ratio to the SRD in the form of a 96% ethanol solution and the SRD of the control group (SRD not supplemented with FaOH and FaDOH) was treated with the same amount of 96% ethanol as previously described [32]. The prepared diets were stored at room temperature, mixed well before use, and used for 1 week before new diets were prepared. The content of FaOH and FaDOH in the rat diets was determined before use and throughout the experiment by liquid chromatography tandem mass spectrometry (LC-MS/MS) [32]. No sign of degradation, oxidation or isomerization of FaOH and FaDOH was observed during the experiments as well as no significant differences in the content of FaOH or FaDOH in the weekly-prepared diet. FaOH and FaDOH that supplemented the SRD were isolated from organic grown carrots of the cultivar Miami with a purity of > 99% by flash column chromatography and preparative high-performance liquid chromatography, and identified by optical rotation, UV-Vis, LC-MS/MS, and NMR spectroscopy as described previously [32].

One hundred and twenty rats were divided into 6 groups of 20 rats receiving different diets in two independent experiments (experiment 1 and 2) plus 24 control rats, with 4 in each experiment not receiving the carcinogen. In experiment 1, which has been described previously [32], the first group of rats received SRD supplemented with 7 µg FaOH and 7 µg FaDOH g^−1^ feed and the other group received SRD not supplemented with FaOH and FaDOH. In experiment 2, four groups of rats received SRD supplemented with 0.16 µg FaOH and 0.16 µg FaDOH g^−1^ feed, 0.48 µg FaOH and 0.48 µg FaDOH g^−1^ feed, 1.4 µg FaOH and 1.4 µg FaDOH g^−1^ feed and 35 µg FaOH and 35 µg FaDOH g^−1^ feed, respectively. The rats started the special diet 14 days before the first AOM injection. The AOM solution was stored for about 1 h at room temperature before being injected. Twenty of the 24 animals in each treatment group were given weekly subcutaneous injections of freshly prepared AOM at a dose of 15 mg kg^−1^ body weight for a period of 2 × 2 weeks separated by a 1-week break. The injection volume used was 0.4 mL at the start and 1.0 mL at the end of the AOM treatments. Four control rats in each treatment group were injected with a volume of sterile 0.9% NaCl related to the body weight.

### 2.3. Autopsy Procedures

After 18 weeks from the first AOM induction, the rats were euthanized and autopsied and examined for macroscopic alterations as previously described [31,32]. Before fixation, the large intestine was evaluated for macroscopic polyp neoplasms, and their diameter and location in the intestine were registered. Biopsies from macroscopic neoplasms > 3 mm were taken and placed on dry ice until stored at −80 °C for gene expression studies. No neoplastic lesions were observed in the four control rats in each treatment group that were only injected with sterile 0.9% NaCl.

### 2.4. Identification and Quantification of Macroscopic Polyp Neoplasms and ACFs

After fixation of the large intestine, the ACFs were visualized by Giemsa stain [6 mL of stock solution (the Central Pharmacy at the Odense University Hospital) in 50 mL of phosphate-buffered saline (PBS), pH 7.2, for 15 min]. Excess stain was rinsed off with PBS. The tissue was placed with the luminal side up in a Petri dish with enough PBS to cover the tissue. The total numbers of ACF and macroscopic polyp lesions for each section were counted independently by two persons blinded to treatment modality, using a stereomicroscope at 40× magnification. Aberrant crypts were distinguished by their increased size and thicker and deeply stained epithelial lining as compared with normal crypts (Figure 2). An ACF may consist of one to several crypts, and in the present study, ACFs were classified as small ACF (1–7 crypts) or large ACF (> 7 crypts), while macroscopic polyp lesions were classified as adenomas (benign tumors > 1 mm). Macroscopic findings were fixed in 4% (*v*/*v*) formaldehyde buffered with 0.075 M sodium phosphate (pH 7), and embedded in paraffin. The tissues from adenomas were cut into 5 µm sections and were stained with hematoxylin and eosin. Additional sections were cut until characterization of the neoplasm was certain.

### 2.5. Immunohistochemical Analysis

Formalin-fixed, paraffin-embedded sections were dewaxed and rehydrated. Antigen retrieval was performed in a boiling water bath with MBO/T-EG buffer for 15 min. After cooling, endogenous peroxidase was blocked with 1.5% (*v*/*v*) H_2_O_2_ for 10 min. Sections were subsequently blocked in PBS + 0.5% (*v*/*v*) goat serum for 30 min. Primary antibodies were incubated overnight at 4 °C in PBS + 0.5% (*v*/*v*) goat serum: rabbit anti-COX-2 (ab19151; abcam, Cambridge, UK) was used at 1:2000. Tissue was illuminated using Dako EnVision+ System- HRP Labelled Polymer Anti-Rabbit (K4003; Dako A/S, Glostrup, Denmark) and chromogen (K3468; Dako A/S, Glostrup, Denmark). Sections were incubated for 10 min. This part of the study was a pilot study and therefore the staining was not quantified.

### 2.6. Gene Expression Study

The gene expression of seven inflammatory markers COX-1, COX-2, NF-κβ, IL-1β, IL-6, TNFα, and peroxisome proliferator-activated receptor-gamma 2 (PPARγ2) in tissue from biopsies from experiment 1 was analyzed by means of real-time quantitative PCR (RT-qPCR). The biopsies included neoplastic tissue from the control group, receiving SRD without the supplement of FaOH and FaDOH and size matched neoplastic tissue from the test group receiving SRD supplemented with 7 µg FaOH and 7 µg FaDOH g^−1^ feed. Prior to the RT-qPCR, RNA from the tissue was extracted using QIAzol and EconoSpin column purification and subsequently converted into complementary DNA (cDNA). Multiplex RT-qPCR was performed using Amplicon master mix on an MYGO mini where the housekeeping gene β-glucuronidase was used as a reference. Each gene was measured in four replicates run pairwise. Primers (MERK) and probes (PentaBase, Odense, Denmark) were reported as Appendix A
Appendix A.

The up- or downregulation of the specific genes was quantified by means of fold changes using the comparative threshold cycle (CT) method as shown in Equation (1) [37,38]. In Equation (1), CT is the threshold cycle, GOI is the gene of interest and IC is the internal control, in this case the housekeeping gene β-glucuronidase, sample A is the treated sample and sample B is the untreated control.
(1)$$2^{-\Delta \Delta CT}=2^{-\left[\left(CT,GOI-CT,IC\right)Sample\;A-\left(CT,GOI-CT,IC\right)Sample\;B\right]}$$

Briefly, by applying Equation (1), the expression of the gene of interest is normalized to an internal control and compared with the untreated control. If the first ΔCT is greater than the second ΔCT, the value of 2^−△△*CT*^ will be less than 1, implying a reduction in the expression, hence a value greater than 1 will indicate an induction in the gene expression. Results are presented as bar charts ± standard deviation (SD).

### 2.7. Statistical Analyses

Statistical analysis of the gene expression data generated by RT-qPCR was performed using SAS JMP Pro 13.0.0 software and the data are presented as mean ± SD and *p* < 0.05 was considered to be significant. Data were analyzed using Student *t* test (two-tailed). Linear regression analysis of ACF data was performed using Stata 15.0 software. Regression analysis of macroscopic polyp neoplasms was determined by Poisson regression, which was performed using Stata 15.0 software. 

## 3. Results

### 3.1. Study of the Effect of FaOH and FaDOH on Colorectal Precancerous Lesions in AOM-Induced Rats

FaOH and FaDOH (Figure 1) were purified (purity > 99%) from extracts of the carrot cultivar Miami by chromatographic methods and identified by spectroscopic and spectrometric techniques. Furthermore, analyses of randomly selected batches of feed of the active arm during the feeding experiments showed no sign of degradation, oxidation or isomerization of FaOH and FaDOH in accordance with the fact that no significant changes in the content of these polyacetylenes were observed during the feeding experiments, which is also in accordance with our previous investigation [32]. Thus, the feed of the active arm contained the prescribed amount of FaOH and FaDOH during the dose-response study (Table 1, Figure 3).

The total number of polyp neoplasms identified under macroscopic examination of the rat intestine were 21 adenomas (> 1 mm) in the group of control rats not receiving FaOH and FaDOH in the diet. This number decreased from 18 adenomas in the treated rats receiving 0.16 µg FaOH and 0.16 µg FaDOH g^−1^ feed to 7 adenomas in rats receiving 35 µg FaOH and 35 µg FaDOH g^−1^ feed, thus indicating a dose-response effect in the development of macroscopic polyp neoplasms (Table 1). The number of adenomas is a count variable with only a mild degree of over-dispersion; hence, Poisson regression was performed on these data and confirmed a dose-response relationship with the number of macroscopic lesions decreasing by 0.167 for each log-fold increase in dose (*p* = 0.007). However, the macroscopic polyp lesions varied in size but the adenomas were generally smaller in the FaOH/FaDOH treated rats compared to the control group (see Appendix A
Appendix A). Adenomas were confirmed as neoplastic by histological analyses. Only a few adenocarcinomas were developed in the observation period.

ACF are clusters of abnormal tube-like glands in the lining of the colon (Figure 2). From clinical intervention studies, most of these crypts are the earliest neoplastic lesion of the colon and only a few are dysplastic or may become dysplastic ACF that eventually may develop into adenomatous polyps [39,40,41,42]. However, in carcinogen-induced rodent models the ACF are preponderantly dysplastic [40,43]. As shown in Table 1, the average number of ACF decreased with increasing dose of FaOH and FaDOH in the diet, and a significant dose-response relationship was observed for both ACF < 7 crypts and ACF > 7 crypts as shown in Figure 3A and Figure 3B, respectively. However, for ACF < 7 crypts, a 5-fold higher dose at the highest doses of FaOH and FaDOH, i.e., from 7 to 35 µg g^−1^ feed did not result in a further reduction in ACF < 7 crypts (Table 1; Figure 3A). For large ACF (> 7 crypts) the effect was also minimal by increasing the concentration of FaOH and FaDOH in the feed at the highest doses of FaOH and FaDOH (Table 1, Figure 3B).

### 3.2. Gene Expression Studies and Immunohistochemical Analysis

The expression level of seven different inflammatory and cancer biomarkers COX-1, COX-2, TNFα, IL-6, NF-κβ, IL-1β, and PPARγ2, were examined in both neoplastic and healthy tissue from biopsies obtained from rats receiving either SRD or SRD supplemented with the two polyacetylenic oxylipins FaOH and FaDOH. The RT-qPCR analyses showed no significant difference in the expression level in healthy tissue for all biomarkers when the rats received FaOH and FaDOH as a food supplement compared to rats receiving only SRD (Figure 4 and Figure 5). However, a significant downregulation in the expression level was detected in five of the seven biomarkers when comparing biopsies of neoplastic tissue from rats receiving a SRD with biopsies of neoplastic tissue from rats receiving SRD supplemented with FaOH and FaDOH. A significant downregulation in the expression of TNFα (Figure 4A), IL-6 (Figure 4B), NF-κβ (Figure 4C), PPARγ2 (Figure 4E) and COX- 2 (Figure 5B) was detected.

Immunohistochemical analysis of COX-2 in normal and neoplastic tissue (Figure 5C−E) confirmed the results of the gene expression study (Figure 5B), although the staining was not quantified. First, an upregulation of COX-2 in neoplastic tissue from rats receiving SRD (Figure 5D) compared to normal colon epithelial tissue (Figure 5C) was observed. Secondly, a clear downregulation of COX-2 in neoplastic tissue from rats receiving SRD supplemented with FaOH and FaDOH (Figure 5E) compared to the control (Figure 5C) was observed.

## 4. Discussion

AOM is a potent carcinogen that has been shown to be an efficient inducer of ACF and other precancerous lesions as well as CRC in rats [32]. Only a small number of early neoplastic ACF lesions will develop to neoplastic polyp lesions and only a small number of these lesions will have the potential to develop into adenomas and cancers [42,44]. In the AOM-induced rat model, the AOM enters the circulation and is metabolized in the liver, and then it is mixed with the feed in the duodenum and transported to the large intestine where the metabolized carcinogen induces neoplastic lesions in the colon of the rats [32]. The cytochrome P450 (CYP) enzymes are a class of heme-containing enzymes involved in phase 1 metabolism of which cytochrome P450 2E1 (CYP2E1) metabolizes many xenobiotics and procarcinogens, and thus they play an important role in the activation of AOM in vivo [45]. Compounds that inhibit or suppress CYP2E1 activity affect AOM metabolism and hence may prevent hypermethylation and carcinogenesis of intestinal cells, and thereby early neoplastic ACF formation as well as other neoplasms [32]. Since administration of FaOH and FaDOH in the present study started two weeks before the first carcinogen dose and continued during the carcinogen administration phase, one mechanism of action for the chemopreventive effect of FaOH and FaDOH could be an inhibition or suppression of AOM metabolism. However, the protective role of these dietary polyacetylenes against xenobiotics and procarcinogens are not due to inhibition or suppression of CYP2E1 activity and other phase 1 enzymes as demonstrated in carbon tetrachloride-induced hepatotoxicity in rodents [46]. In fact, FaOH and FaDOH seem to exert their chemopreventive protection against carcinogens and other toxic compounds through the activation of the Kelch-like ECH-associated protein 1 (Keap1)/nuclear factor [erythroid-derived 2]-like 2 (Nrf2)/antioxidant response element (ARE) pathway due to their S-alkylating properties of cysteine, thereby inducing the formation of antioxidant and other cytoprotective phase 2 enzymes [47,48,49]. The alkylating properties of these dietary polyacetylenes may not only explain their ability to protect against the formation of carcinogens through the Keap1/Nrf2/ARE pathway but also their role in the inhibition of inflammatory markers of the NF-κB signaling pathway. The latter primarily explain the effect of FaOH and FaDOH on neoplastic lesions and their effect on the development of CRC as discussed below. Furthermore, as described previously, FaOH and FaDOH exhibit a preventive effect on cancer cells grown in vitro. Still, it could be interesting in future investigations to determine more precisely the importance of FaOH and FaDOH in the detoxification of chemical carcinogens via the Keap1/Nrf2/ARE pathway and their overall cancer preventive role. 

A reduction in the growth rate of early neoplastic ACF’s and neoplastic polyp lesions is effected after the administration of AOM, when it is no longer present in the rats. Hence, the cancer primed rat model used in this study makes it possible not only to evaluate the chemopreventive effect of FaOH and FaDOH during the carcinogenesis but also their potential as inhibitors of tumor growth. The doses of FaOH and FaDOH used in the rat feed in the present dose-response study corresponds to a realistic daily human intake of carrots, except maybe for the highest concentration of 35 µg FaOH and 35 µg FaDOH g^−1^ feed [32]. The results of this study may therefore be used to design clinical experiments with, for example, food products to determine a possible preventive effect of a dietary intake of FaOH and FaDOH.

The literature on the clinical significance of ACF as precursors of colorectal adenomas and CRC is modest and because ACF are the earliest precursor lesions in colorectal carcinogenesis, it is not likely that all early neoplastic ACF can be used as surrogate markers in CRC chemoprevention trials in humans [40,50]. However, in carcinogen-induced rodent carcinogenesis models the frequency of dysplastic ACF is much higher compared to humans, and if dysplastic ACF are precursors of colorectal adenomas or CRC, then the ACF number may be a useful tool to identify promising chemopreventive agents in cancer primed rodent models [43,50,51]. In the present investigation, a correlation appeared to exist between the number of early neoplastic ACFs and the number of neoplastic polyp lesions. The average number of ACF decreased with an increasing dose of FaOH and FaDOH in the diet, and a significant dose-response relation was observed for both ACF < 7 crypts and ACF > 7 crypts, although a further inhibitory effect on the formation of ACFs at the highest doses of FaOH and FaDOH was minimal. On the other hand, the largest effect on the total number of macroscopic polyp neoplasms was observed at the highest doses of FaOH and FaDOH in the feed, which indicates that the anti-neoplastic effect is most significant at these doses. Thus, it appears that an optimal anti-neoplastic effect is achieved with doses between 7–35 µg of FaOH and FaDOH g^−1^ feed in the cancer primed rat model used in this study. Interestingly, adenomas were in general smaller in the FaOH and FaDOH treated rats, especially at the highest doses, i.e., > 1.4 µg of FaOH and FaDOH g^−1^ feed, compared to the control group. This is also in accordance with our previous findings [32]. These results not only demonstrate a clear dose-dependent chemopreventive effect of FaOH and FaDOH on the formation of colorectal early neoplastic ACF and neoplastic polyp lesions, but also indicate that these dietary polyacetylenic oxylipins may to some extent exert a growth inhibition of neoplastic lesions, as they not only reduce the number of macroscopic polyp neoplasms but also their size.

Chronic inflammation is believed to play an important role in the early stage of malignant transformation, including CRC [52]. The use of NSAIDs, such as aspirin, reduces the overall number and size of adenomas in patients by inhibition of COX-1 and COX-2 [53]. Furthermore, healthy individuals using NSAIDs regularly reduce their risk of developing colorectal cancers by 40% to 50% [12]. Indeed, the accumulating clinical and experimental evidence now supports a potent anti-tumorigenic efficacy of inhibiting COX-2 with NSAIDs as a chemotherapeutic strategy for CRC [10]. From studies in cell cultures, it is known that FaOH and FaDOH are able to inhibit pro-inflammatory cytokines and enzymes such as COX, as described in the introduction. These downstream inflammatory markers therefore seem to constitute a likely explanation for the anti-neoplastic effects of FaOH and FaDOH. However, it cannot be excluded that FaOH and FaDOH have an inhibitory effect on other inflammatory markers not included in the present investigation. Gene expression studies and immunohistochemical analysis showed, as expected, an upregulation of COX-2 in the neoplastic tissue [10] whereas COX-1 was not significantly affected. However, a significant downregulation of COX-2 in adenomas from rats receiving SRD supplemented with FaOH and FaDOH compared to adenomas of the control group, implicate the involvement of COX-2 in CRC development and as an important target for the chemopreventive effects of these polyacetylenes. Based on the results of this study, it appears that FaOH and FaDOH in combination act as selective COX-2 inhibitors.

The NF-κB signaling pathway regulates the immune response and inflammation, which has also been implicated in carcinogenesis and is crucial for neoplastic transformation and promotion as it is the main regulator of the pro-inflammatory cascade, including the expression and formation of COX-2 [54]. As expected, the gene expression of NF-κB was significantly downregulated in tumor tissue in rats receiving SRD supplemented with FaOH and FaDOH compared to the control group that only received the SRD. This further confirms that inflammation plays a key role in early neoplastic formation and that the chemopreventive effect of FaOH and FaDOH in the colon is linked to their anti-inflammatory activity. Further evidence for the anti-inflammatory action of FaOH and FaDOH was confirmed by the gene expression of TNFα and IL-6, which are members of the pro-inflammatory cytokine cascade and have been implicated in carcinogenesis [55,56,57]. Increased TNFα levels are linked to increased leukocyte infiltration and tumor formation and upregulation of TNFα levels are usually detected in colorectal neoplasms, and in animal models of CRC [55,56]. However, gene expression of TNFα was significantly downregulated in rats receiving SRD supplemented with FaOH and FaDOH compared to the control group, which indicates that these polyacetylenes exert an effect on the regulation of this pro-inflammatory biomarker. IL-6 is highly upregulated in many cancers and is considered as one of the most important pro-inflammatory cytokines during tumorigenesis and metastasis [57]. IL-6 was significantly downregulated in rats receiving FaOH and FaDOH in the diet compared to the control group, which again confirmed the anti-inflammatory effect of FaOH and FaDOH in the NF-κB signaling pathway, and thus the downregulation of COX-2 formation.

Epithelial IL-1β is known to be regulated by NF-κB but may also be induced by DNA damage via a NF-κB-independent mechanism leading to mucosal inflammation in the gut [58]. In the present study, FaOH and FaDOH showed no effect on the gene expression of IL-1β in neither healthy nor tumor tissue, which indicates that cytokines such as TNFα and IL-6, rather than IL-1β, play a role in the regulation of COX-2 in neoplastic tissue in this rat model.

PPARγ is an essential nuclear receptor controlling the expression of a large number of regulatory genes in lipid metabolism, insulin sensitization, inflammation, and in cell proliferation. The isoform PPARγ2 is mostly found in adipose tissue and the intestine and is highly expressed in these tissues [59]. In the colon, PPARγ play a key role in the control of intestinal inflammation such as ulcerative colitis [59]. In addition, PPARγ has been found to have an anti-neoplastic property as it can induce apoptosis and differentiation of colon cancer cells both in vivo and in vitro [60]. Thus, PPARγ could play a significant role in the development of CRC. The regulation of PPARγ expression in the colon is unresolved but PPARγ expression may be upregulated by intestinal-microbial interactions involving lipopolysaccharides of Gram-negative bacteria and/or ligands of PPARγ [60]. In the present study, PPARγ expression was significantly downregulated in rats receiving FaOH and FaDOH in the diet compared to the control group, which indicates that PPARγ is not an important target for the anti-neoplastic effect of these polyacetylenic oxylipins. FaOH and FaDOH have previously been shown to act as partial PPARγ agonists, which seems to explain their anti-diabetic properties [22]; thus, an increased expression of PPARγ in epithelial cells would have been expected in rats receiving FaOH and FaDOH in the diet. However, recent research has also shown that FaOH and FaDOH alter the gut microbiota composition in the AOM-induced rat model [61], which could be the main reason for the downregulated expression of PPARγ observed in rats receiving FaOH and FaDOH in the diet.

## 5. Conclusions

Collectively, the results of the present study show that FaOH and FaDOH in combination have a dose-dependent chemopreventive effect on colorectal neoplastic lesions in a cancer primed rat model, and that this effect is most likely due to inhibition of downstream inflammatory markers in the NF-κB signaling pathway. In particular, COX-2 seems to be an important target for the anti-inflammatory effect of FaOH and FaDOH, although it may not be the only downstream inflammatory marker contributing to the chemopreventive effect of these dietary polyacetylenes. The present study suggests the need for precise dietary advice to prevent CRC, and to develop new selectively COX-2 inhibitors with no or only minor side effects. 

## Figures and Tables

**Figure 1 nutrients-11-02223-f001:**
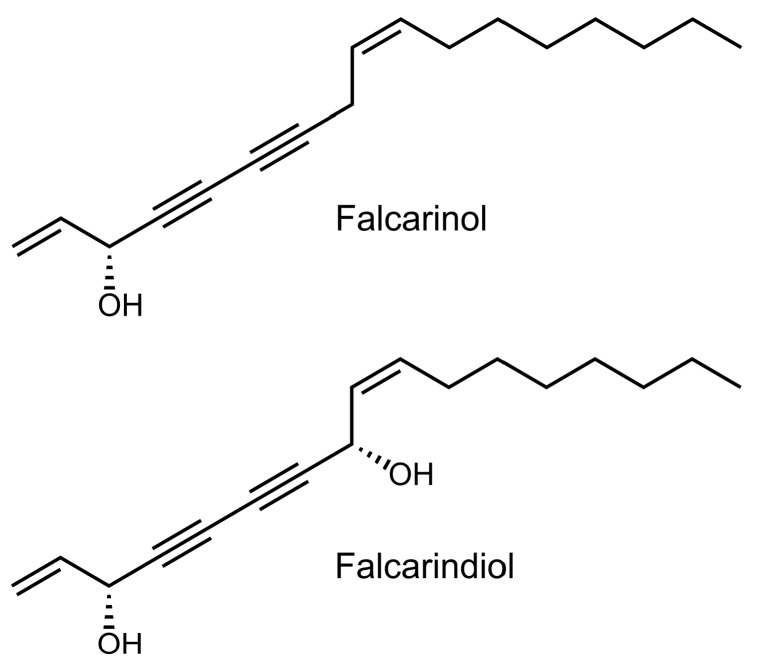
Chemical structures of (3*R*,9*Z*)-1,9-heptadecadiene-4,6-diyne-3-ol (falcarinol; FaOH), and (3*R*,8*S*,9*Z*)-1,9-heptadecadiene-4,6-diyne-3,8-diol (falcarindiol, FaDOH) tested in colorectal cancer (CRC) primed rats.

**Figure 2 nutrients-11-02223-f002:**
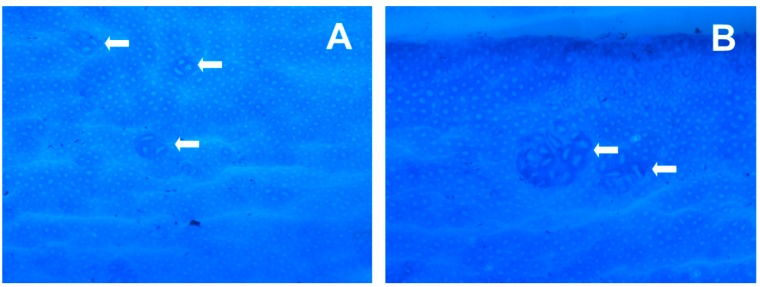
Appearance of small and large aberrant crypt foci (ACF) in rat colon tissue. (**A**) Small ACF (enlarged) < 7, where white arrows show crypts of 3 and 4. (**B**) Large ACF (enlarged) > 7, where white arrows show crypts > 7. The epithelial lining was visualized by Giemsa stain (×200).

**Figure 3 nutrients-11-02223-f003:**
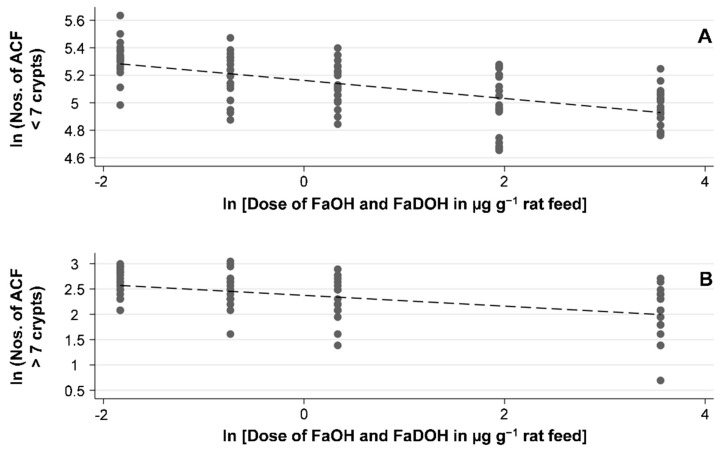
FaOH and FaDOH in the diet of AOM-challenged rats show a significant dose-response effect with regard to different sizes of early neoplastic lesions of ACF in a natural logarithmic (ln) scale. (**A**) Linear regression of the ln average numbers of ACF crypts < 7 as a function of the ln dose of FaOH and FaDOH in µg g^−1^ rat feed show a significant linear correlation (*R^2^* = 0.3742, *p* < 0.001). (**B**) Linear regression of the ln average numbers of ACF crypts > 7 as a function of the ln dose of FaOH and FaDOH in µg g^−1^ rat feed show a significant linear correlation (*R^2^* = 0.2451, *p* < 0.001).

**Figure 4 nutrients-11-02223-f004:**
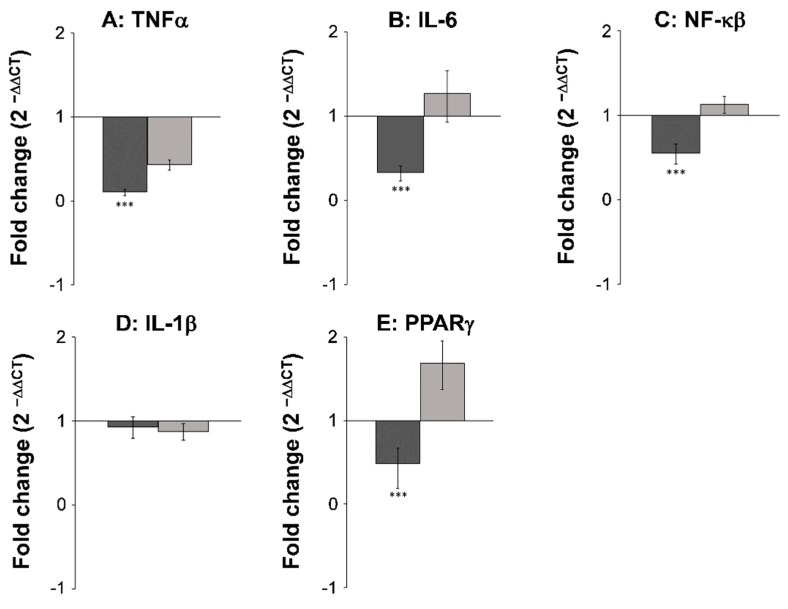
FaOH and FaDOH downregulates expression of TNFα, IL-6, NF-κβ and PPARγ2 in neoplastic tissue in colorectal cancer primed rats. Up- and downregulation of TNFα (**A**), IL-6 (**B**), NF-κβ (**C**), IL-1β (**D**) and PPARγ2 (**E**) in neoplastic and healthy tissue in rats after treatment with FaOH and FaDOH. The up- and downregulation of the above-mentioned biomarkers was calculated using the CT method. Dark grey bars represent neoplastic tissue and light grey bars represent healthy tissue. All values represent the mean ± SD. *** *p* < 0.001.

**Figure 5 nutrients-11-02223-f005:**
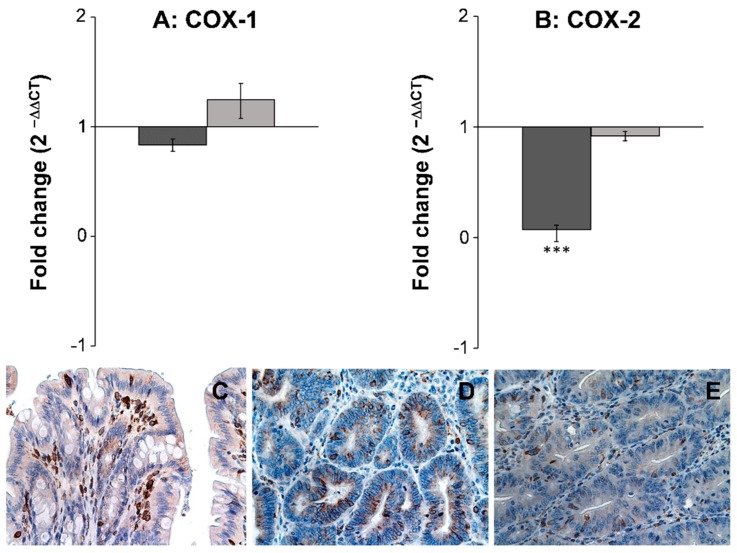
FaOH and FaDOH downregulates expression of COX-2 in neoplastic tissue in colorectal cancer primed rats. (**A**) Gene expression of COX-1 in neoplastic and healthy tissue after treatment with FaOH and FaDOH. (**B**) Gene expression of COX-2 in neoplastic and healthy tissue after treatment with FaOH and FaDOH. Up- and downregulation of COX-1 and COX-2 was calculated using the comparative CT method. Dark grey bars represent neoplastic tissue, light grey bars represent healthy tissue. All values represent the mean ± SD. *** *p* < 0.001. (**C**) Immunohistochemical analyses of COX-2 in normal rat colon epithelial tissue (control), where cells stained positive for COX-2 enzyme (red-brown color) mainly consist of immune infiltrating cells. (**D**) Neoplastic tissue from rats receiving SRD. (**E**) Neoplastic tissue from rats receiving SRD supplemented with FaOH and FaDOH.

**Table 1 nutrients-11-02223-t001:** The mean ± SD of small ACF (<7 crypts) and large ACF (>7 crypts) and the total number of macroscopic polyp neoplasms (benign tumors > 1 mm) in 6 groups of 20 azoxymethane (AOM)-induced rats receiving a standard rat diet (SRD) or a SRD supplemented with different doses of FaOH and FaDOH. − Indicate no data are available.

Size of Neoplasms	µg FaOH g^−1^ Feed and µg FaDOH g^−1^ Feed					
	0 (*n* = 20)	0.16 (*n* = 20)	0.48 (*n* = 20)	1.4 (*n* = 20)	7 (*n* = 20)	35 (*n* = 20)
Mean ACF < 7 crypts	205 ± 36	207 ± 28	180 ± 29	171 ± 26	150 ± 31	145 ± 19
Mean ACF > 7 crypts	−	14 ± 3.7	12 ± 4.1	10 ± 3.7	−	8 ± 3.5
Total number of macroscopic polyp neoplasms	21	18	19	13	12	7

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
