# Peer review of "Dietary Polyacetylenic Oxylipins Falcarinol and Falcarindiol Prevent Inflammation and Colorectal Neoplastic Transformation: A Mechanistic and Dose-Response Study in A Rat Model"

_nutrients, 2019, doi:10.3390/nu11092223_

Round 1

Reviewer 1 Report

The manuscript titled “Dietary polyacetylenic oxylipins falcarinol and falcarindiol prevent inflammation and colorectal neoplastic transformation through COX-2 inhibition: A mechanistic and dose-response study in a rat model” focuses on the role of anti-inflammatory polyacetylenic oxylipins, falcarinol (FAOH) and falcarinol (FaDOH) found in apiaceous vegetables in the prevention of early stage colorectal cancer development. The authors performed in-vivo study and showed that FAOH and FaDOH have inhibitory effects on key biomarkers in the NF-κB signaling pathway, particularly the downstream target COX-2 and prevent the development of neoplastic precursor lesions in intestine. With these findings the authors suggest FaOH and FaDOH in combination has a dose-dependent chemo-preventive effect on colorectal neoplastic lesions. The manuscript is well-written and would be interesting to the intended audience.

Author Response

No comment - we have revised the manuscript to clarify the conclusions.

Reviewer 2 Report

Manuscript by Morten Kobaek-Larsen et. al. investigated effect of Falcarinol and Falcarindiol on Azoxymethane (AOM) induced carcinogenesis in rat model. Abberant Crypt Foci (ACF) and adenomas were isolated and several inflammatory markers including Cox-1 and Cox-2 were assessed.

There are some weaknesses to be resolved.

Minor:

Abstract is incomplete and not to the point.

Major:

There are no experiments to justify the conclusion that suppression of inflammatory markers are due to the Cox-2 inhibition. Thus title may need to change.

RTPCR mRNA analyses are not enough to show the suppression of the chosen genes. Authors should also show western blots.

Cox enzyme is not the only downstream target of the chemopreventive FaOH and FaDOH containing diet. There could be many other downstream major pathways that may in turn influence inflammatory markers. In other words such studies are not completed till we assess cell culture models, which is probably beyond the scope of this investigation.

There are many reports that suppression of Cox-2 leads to the inhibition of inflammatory markers. However, effect of FaOH and FaDOH is not investigated in cell culture models.

Author Response

We thank the reviewer for the constructive comments to our manuscript and based on these comments we have made the following changes in the manuscript.

Minor points:

“Abstract is incomplete and not to the point.”

Response: We have followed the guideline of Nutrients for preparing the Abstract, which should include background, purpose, methods, results and conclusion. We admit the Abstract could be improved and we have there revised the Abstract so it is more to the point.

Major points:

“There are no experiments to justify the conclusion that suppression of inflammatory markers are due to the Cox-2 inhibition. Thus title may need to change.”

Response: We agree that we have not provided solid evidence that COX-2 inhibition is the main factor for the anti-inflammatory and anti-neoplastic effect of FaOH and FaDOH, although COX-2 play a major role in the development of colorectal cancer as also explained in the paper. We have therefore changed the title of the manuscript by deleting “through COX-2 inhibition” in the title.

“RTPCR mRNA analyses are not enough to show the suppression of the chosen genes. Authors should also show western blots.”

Response: We thank the reviewer for this comment and agree it could have strengthened the study to do western blots as follow up analysis. We believe, however, that we can conclude which genes are upregulated based on the RT-qPCR studies alone as this is a well established technique used to determine up- and downregulation of genes. Furthermore, the down-regulation has been shown in immunohistochemical analysis of contents of COX-2 in same tissue samples as analysed by RT-qPCR.

“Cox enzyme is not the only downstream target of the chemopreventive FaOH and FaDOH containing diet. There could be many other downstream major pathways that may in turn influence inflammatory markers. In other words such studies are not completed till we assess cell culture models, which is probably beyond the scope of this investigation.”

Response: We agree with the reviewer that their might be many other downstream targets than those investigated in the present investigation, including COX enzymes that could be important in order to explain the chemopreventive effects of FaOH and FaDOH. However, we have tried to justify our choice of inflammatory markers in the present investigation based on the anti-inflammatory effect of FaOH and FaDOH in various cell cultures. To include other possible inflammatory markers that could be inhibited by FaOH and FaDOH and thus may play a role in their chemopreventive effects would require extensive studies in cell culture models, which is beyond the scope of this investigation as also described in the Discussion, line 379-385 in the revised manuscript.

“There are many reports that suppression of Cox-2 leads to the inhibition of inflammatory markers. However, effect of FaOH and FaDOH is not investigated in cell culture models”

Response: It is correct that we have not investigated FaOH and FaDOH in cell culture models in the present investigation. However, the inhibitory effect of FaOH and FaDOH on various inflammatory markers, including COX has previously been investigated in cell culture models as also described in the introduction and also explained at line 379-380 in the Discussion in the revised manuscript. We have marked the text in yellow in the Introduction and Discussion to point out the text describing polyacetylenes, COX and cell cultures.

We thank the reviewer for the valuable comments, which have helped us to improve the manuscript significantly.

Reviewer 3 Report

The current study examines the effects of falcarinol (FaOH) and Falcarindiol (FaDOH) in chemoprevention of colorectal cancer.  Using the AOM-induced model of colon carcinogenesis in rats, the authors show that pre-treatment with increasing doses of a 1:1 ration of FaOH and FaDOH in standard rat diet prior to administration of AOM, followed by continued exposure to the modified diet resulted in a significant dose dependent reduction in the development of aberrant crypt foci and adenomas in rats.  Further, since the doses used were mostly equivalent to that of human dietary consumption of relevant vegetables, the results could be useful in the prevention of colorectal cancer in humans.

The study is straightforward and well designed.  However, the scope of the studies to understand the underlying mechanisms is very limited, and only examines genes that have already been previously established to be involved in inflammation driven CRC, and therefore do not add novel findings to what is already known in literature. Nevertheless, the conclusions are supported by the results in the study, thus it possible to publish the findings with respect to the mechanisms by which FaOH and FaDOH may function to reduce the incidence of CRC in rats.